# Accumulation of 4-Hydroxynonenal Characterizes Diabetic Fat and Modulates Adipogenic Differentiation of Adipose Precursor Cells

**DOI:** 10.3390/ijms242316645

**Published:** 2023-11-23

**Authors:** Giuseppe Murdolo, Desirée Bartolini, Cristina Tortoioli, Cristiana Vermigli, Marta Piroddi, Francesco Galli

**Affiliations:** 1Department of Internal Medicine, Endocrinology and Metabolism, Azienda Ospedaliera S. Maria Misericordia, University of Perugia, Piazzale Gambuli, I-06081 Perugia, Italycri.vermigli@gmail.com (C.V.); 2Department of Pharmaceutical Sciences, Section of Applied Biochemistry and Nutritional Sciences, University of Perugia, I-06081 Perugia, Italyfrancesco.galli@unipg.it (F.G.); 3Molecular Horizon srl, I-06084 Bettona, Italy

**Keywords:** adipose precursor cells, insulin resistance, obesity, oxidative stress, 4-hydroxynonenals

## Abstract

Redox imbalance in fat tissue appears to be causative of impaired glucose homeostasis. This “proof-of-concept” study investigated whether the peroxidation by-product of polyunsaturated n-6 fatty acids, namely 4-hydroxynonenal (4-HNE), is formed by, and accumulates in, the adipose tissue (AT) of obese patients with type 2 diabetes (OBT2D) as compared with lean, nondiabetic control subjects (CTRL). Moreover, we studied the effects of 4-HNE on the cell viability and adipogenic differentiation of adipose-derived stem cells (ASCs). Protein–HNE adducts in subcutaneous abdominal AT (SCAAT) biopsies from seven OBT2D and seven CTRL subjects were assessed using Western blot. The effects of 4-HNE were then studied in primary cultures of ASCs, focusing on cell viability, adipogenic differentiation, and the “canonical” Wnt and MAPK signaling pathways. When compared with the controls, the OBT2D patients displayed increased HNE–protein adducts in the SCAAT. The exposure of ASCs to 4-HNE fostered ROS production and led to a time- and concentration-dependent decrease in cell viability. Notably, at concentrations that did not affect cell viability (1 μM), 4-HNE hampered adipogenic ASCs’ differentiation through a timely-regulated activation of the Wnt/β-catenin, p38MAPK, ERK1/2- and JNK-mediated pathways. These “hypothesis-generating” data suggest that the increased accumulation of 4-HNE in the SCAAT of obese patients with type 2 diabetes may detrimentally affect adipose precursor cell differentiation, possibly contributing to the obesity-associated derangement of glucose homeostasis.

## 1. Introduction

Obesity and type 2 diabetes (T2D) are closely linked as dire threats to public health. Although the relationship between weight gain and the development of T2D is widely established, compelling evidence indicates that obesity is not synonymous with metabolic dysfunctions [1]. Indeed, rather than fat mass expansion, per se, dysregulated fat, characterized by reduced adipogenesis with enlarged adipose cells (hypertrophic obesity) in the subcutaneous (sc) abdominal adipose tissue (AT), appears to be a key determinant of the obesity-associated derangement of glucose homeostasis [2]. The impaired ability to recruit and differentiate new fat precursor cells into the adipogenic lineage limits the capability to store excess energy in mature adipose cells, leading to expanded adipose cells, ectopic lipid deposition (i.e., lipotoxicity) and, ultimately, the impairment of glucose control [3,4]. Therefore, the basic pathways involved in the commitment of multipotent adipose stem cells (ASCs) to the adipogenic lineage “in vivo” has attracted a great deal of interest [3,5,6]. In this context, the “canonical” wingless-type mouse mammary tumor virus integration site family (*Wnt*) has a central role in the regulation of the metabolic reprogramming of ASCs [3,5,6]. Basically, the *Wnt* pathway is highly active in undifferentiated ASCs [3,7,8,9], and terminating the *Wnt* signal is a prerequisite that allows adipogenic differentiation [10]. It has thus been postulated that the aberrant activation of *Wnt*/β-catenin signaling in ASCs may be an early instigator of an unhealthy metabolic phenotype [11,12,13]. The regulation of *Wnt*/β-catenin signaling remains complex, since several cues may differently modulate this pathway [14].

Novel insights into this topic were recently gained through the characterization of lipid peroxidation by-products. Basically, lipid peroxidation refers to the oxidative degradation of lipids with the consequent formation of lipid peroxides. AT is the main storage compartment of ready oxidizable lipids and thus appears to be highly susceptible to oxidative stress and lipid peroxidation. These by-products, in turn, may either induce a variety of toxic effects or act as “signaling mediators” involved in cell signal transduction and the regulation of gene expression [15,16]. We have previously demonstrated that human ASCs appear to be very sensitive and vulnerable to different cholesterol autoxidation products (i.e., oxysterols), which hamper adipogenic differentiation and contribute to fat dysregulation [17]. These data are in accordance with the concept that impaired redox homeostasis in accumulated fat may link the hampered adipogenic potential of ASCs with obesity-associated metabolic dysfunctions [18,19,20,21,22].

In this scenario, 4-hydroxy-2-nonenal (4-HNE), a major nonenzymatic oxidation product of membrane lipids containing polyunsaturated n-6 acyl groups, provides further molecular insights for the importance of oxidative stress in adipose biology. Several studies have drawn correlations between increased 4-HNE formation and the development of insulin resistance (IR) in skeletal muscle and adipose cells [23,24,25,26,27]. Moreover, 4-HNE exposure led to the activation of the canonical *Wnt* pathway in different cell lines [28]. Yet, the occurrence of lipoxidation in human obesity and the effects of 4-hydroxyalkenals in ASCs at the cellular and molecular levels remain poorly explored.

Based on the forgoing, the objectives of the present study were as follows: (1) to test the hypothesis whether 4-HNE is detected in human adipose tissue and accumulates in the fat of obese patients with type 2 diabetes as compared with lean, nondiabetic controls; (2) to evaluate the role of this aldehyde product in adipose precursor cell viability and adipogenic differentiation; and (3) to address the signaling pathways underlying the mechanistic effects of 4-HNE on adipogenesis.

## 2. Results

### 2.1. Subjects Characteristics and Protein–HNE Levels in Adipose Tissue

The characteristics of the study subjects are outlined in Table 1. According to the inclusion criteria, the OBT2D patients were treated with metformin and oral hypoglycemic agents. By definition, the OBT2D patients displayed a higher BMI, waist circumference, WHR, and whole-body and abdominal fat, as well as higher HbA1c and a larger adipose cell size (105 ± 1 vs. 70.5 ± 3.2 µm; *p* < 0.001), when compared with the CTRL individuals.

Notably, protein–HNE levels in AT of OBT2D patients were all significantly higher (almost 2.5-times) in comparison with those observed in the CTRL counterparts (*p* < 0.001) (Figure 1). SDS-PAGE analysis revealed increased abundance of adducts on a subset of proteins of different sizes ranging from ≈20 to ≈90 kDa. Interestingly, multivariate analysis (Table 2) indicated that the cell size of the donor, along with age and sex, explained about 60% of the adipose 4-HNE variance (model R^2^ adj. 0.622; *p* < 0.001). However, only adipocyte diameter was a significant and strong predictor of the adipose 4-HNE accumulation (standardized regression coefficient β = 0.807; *p* < 0.001), independently from age and sex (as potential confounding variables).

#### 2.1.1. Effects of 4-HNE on ASC Viability and Differentiation

As long as exposure to increased adipose 4-HNE accumulation in OBT2D may have damaged and “committed” ASCs to the altered response, the effects of 4-HNE on ASC viability and differentiation were studied occurred in cells isolated from the CTRL individuals (n = 4 subjects).

When increasing doses (0.05–50 µM) of 4-HNE were separately analyzed, a different concentration- and time-sensitive decrease in cell viability was seen (Figure 2). In a concentration range of 0.05–10 µM, treatment with 4-HNE did not affect cell viability, which, in turn, was significantly decreased (cell viability ~10% in cells treated with 4-HNE vs. CTRL) after exposure to higher 4-HNE concentrations (>10 µM). Notably, at each time point, the IC50 of 4-HNE was roughly similar (~20 µM). Finally, at low concentrations (1 µM), 4-HNE exposure did not significantly affect cell viability, and this was also the case after three weeks of starvation (cell viability ~80% in cells treated with 4-HNE vs. CTRL).

Based on these data, in order to avoid “artifactual” findings on differentiation due to a reduced cell number, the effect of 4-HNE on ASCs’ adipogenic differentiation was studied using a final concentration of 1 µM of this compound in the culture medium. As compared to the control (treatment with the 4-HNE vehicle DMSO), treatment with 1 µM 4-HNE inhibited the adipogenic differentiation of uncommitted ASCs, as ascertained by either contrast phase microscopy or Oil Red O staining by day 14 after the addition of a supplementation medium (Figure 3).

#### 2.1.2. ROS Production and the Wnt/β-Catenin and MAPK Signaling Pathways

To better ascertain the molecular mechanisms underlying the effects of 4-HNE in ASCs, we first examined the significance of ROS. As partly expected, the incubation of progenitor cells with 4-HNE induced a dose-dependent burst of ROS, which reached significant values only when high (“toxic”) concentrations (50 µM) were tested (Appendix A).

We then asked whether 4-HNE (1 µM concentration) may directly trans-activate the Wnt/β-catenin and the MAPK pathways. To address the recruitment of Wnt, we first examined the effect on cellular β-catenin and its phosphorylation at threonine 41 (pThr-41) and serine 33 and 37 (pSer-33/37) (Figure 4; panels A and B). The hyperphosphorylation of β-catenin by GSK3β at Thr-41 and Ser-33/37 primes β-catenin for ubiquitination and subsequent proteasomal degradation. Notably, after 24 h of exposure, 4-HNE transiently increased β-catenin phosphorylation, the levels of which declined after 48 and 72 h. Accordingly, at the same time point, we observed an up-regulation of total β-catenin, accompanied by a down-regulation of axin and the recruitment of GSK3β, proteins that cooperate in preventing β-catenin degradation in the destruction complex.

Finally, we determined whether 4-HNE may also affect mitogen-activated protein kinases (MAPKs). Interestingly, 4-HNE induced an early activation (24 h) of the extracellular-signal-regulated kinase (ERK 1/2) (Figure 5, panel A), the c-Jun N-terminal kinase (SAPK/JNK) (Figure 6, panel A) and p38 (Figure 7, panel A) signaling pathways, as ascertained by the increased ratios of the phosphorylated protein to the total protein and/or to the ratio of the housekeeping protein signals. In comparison with the 24 h time point, after 72 h of challenge with 4-HNE, SAPK/JNK phosphorylation (Figure 6, panel C) was blunted, while the up-regulation of p38 (Figure 7, panel C) and ERK1/2 (Figure 5, panel C) remained similar and still significant as compared with the control, respectively.

## 3. Discussion

The results of this proof-of-concept study demonstrate that the accumulation of protein HNE–adducts characterizes human “diabetic” fat. Moreover, we showed that 4-HNE hampers the adipogenic potential of human adipose progenitor cells through the time-regulated activation of the *Wnt* and MAPK signaling pathways.

Adipose tissue holds the key to improving our understanding of systemic metabolic homeostasis [29]. Redox imbalance in expanded fat has been consistently associated with adipose dysfunction and obesity-linked metabolic perturbations [19]. The first finding of our study, that is, the identification of HNE–protein adducts in AT, supports the concept of the “in vivo” formation of toxic lipid aldehydes in human fat [26]. This information appears to be of importance with respect to clinical readouts for the unhealthy metabolic outcomes of hypertrophic obesity [2,4]. From a mechanistic standpoint, protein–HNE modifications (i.e., protein carbonylation) are considered as relatively long-lived “footprints” of lipid peroxidation [30]. The biological detection of specific and metastable protein–aldehyde adducts in the adipose microenvironment makes these products a useful “in vivo” biomarker of oxidative stress. In line with such an assumption, our multiple regression analysis (Table 2) showed that the increased accumulation of protein–HNE adducts that we found in the “diabetic” fat tissue was strongly associated with, and explained by, the larger diameter of the adipose cells, a reliable marker of restricted adipogenesis and systemic insulin resistance [2,31]. Our data are partly in line with those reported by Jaganjac et al. in regard to omental fat depots [32]. In contrast, Jankovic et al. showed a roughly similar expression of 4-HNE by comparing the subcutaneous fat of obese “metabolically healthy” and “metabolically obese” premenopausal women [33], a discrepancy which can be partly explained by differences in the clinical characteristics of the study subjects, study methodology, and techniques (i.e., sample preparation). However, our findings are in accordance with the concept that increased protein carbonylation also occurs in the subcutaneous abdominal fat depot, the primary site for lipid accumulation in the body, providing a possible mechanistic link between adipose lipoxidation and the development of insulin resistance, as previously postulated [34,35]. On the other hand, both increased peroxidation and/or hindered neutralizing (i.e., lipid antioxidant) pathways may well explain the differing abundance of 4-HNE found in the AT of obese patients, suggesting a defective turnover of this protein as a post-translational modification induced by the metabolic effects of insulin resistance [23]. Finally, since a defective metabolism of this aldehyde adduct may represent the “snapshot” of a long-term molecular alteration process that precedes the onset of adipose dysfunction, a different duration of exposure to 4-HNE in obese fat, compared to healthy control fat, cannot be ruled out [33]. Whatever the mechanisms, intracellular 4-HNE accumulation in fat may strongly modify the local microenvironment by altering multiple aspects of adipose biology [34,35].

As far as restricted adipogenesis in SCAAT promotes insulin resistance [2,4], we hypothesized that 4-HNE might act as a molecular instigator of blunted ASC differentiation. The literature provides contradictory findings on the effects of hydroxyalchenals on adipogenesis. Recurring biases in this regard concern the selected cell type (i.e., murine or human cell lines), as well as the exposure time and concentrations of 4-HNE used for the experiments [36]. In the current study, we demonstrated that acute as well as prolonged 4-HNE exposure detrimentally affects the adipogenic potential of ASCs. Indeed, a short-term treatment with 4-HNE induced a burst of ROS formation, which actually occurred only when high (not physiological) and “toxic” concentrations were tested (50 µM). This observation lays the framework for a model in which “feed forward” cascades of 4-HNE generation and ROS formation likely occur within the peroxidative challenge to dysfunctional fat [34]. We thus hypothesize that under conditions of a positive energy balance (i.e., overfeeding), the accumulation of 4-HNE in adipocytes may offer a defense mechanism for trapping harmful lipoxidation products in protein components for subsequent homoeostatic activity carried out by the cellular “*proteostasis*” system. However, to prevent further fat mass expansion, an excess of 4-HNE in hypertrophic adipocytes further enhances redox unbalance and blunts adipogenic differentiation, leading, ultimately, to unfavorable metabolic outcomes.

In this regard, an interesting finding of the present study was the different concentration- and time-dependent “susceptibility” of ASCs to 4-HNE exposure (Figure 3). At the cellular level, we showed that while increasing (and not physiologic) doses of 4-HNE behave as a cytotoxic signal, low 4-HNE concentrations (i.e., 0.05–10 µM) did not significantly affect adipose precursor cell viability. These data suggest a unique dual and “hormetic” action of this compound on human ASCs [33,37]. Interestingly, not all cell types are equally susceptible to 4-hydroxyalkenal-induced damage [36]. We have previously reported that ASCs appear to be vulnerable to different cholesterol autoxidation products, which, at concentrations resembling those found “in vivo” (1 μM), similarly hamper adipogenic differentiation with negligible effects on cell vitality [17]. These effects of oxysterols partly coincide with those found in the present study for the activity of “physiologic” concentrations (1 μM) of 4-HNE, which, despite an apparently higher IC50 in ASC viability as compared to oxysterols [17], similarly impaired the adipogenic potential of precursor cells. It is worth noting that the lack of an effect on cell viability after the exposure of ASCs to the 1 μM dose of 4-HNE provided a solid experimental background for testing adipogenic differentiation, avoiding artifactual findings due to a reduced cell number. Together, these results underscore the inherent and possibly synergistic mechanisms of damage to human ASCs induced by individual peroxidation products that are worth investigating in ongoing studies.

To better ascertain the signaling pathways through which 4-HNE affects ASC adipogenic commitment and differentiation, we focused on both the canonical *Wnt* and the MAPK pathways [38]. As long as exposure to increased 4-HNE levels in the AT of OBT2D damaged and “committed” resident precursor cells, the effects of 4-HNE on adipogenesis were studied in ASCs isolated from CTRL individuals. Notably, the temporal expression profiling experiments demonstrated, for the first time, a unique “dual action” of 4-HNE on *Wnt* signaling: after 24 h incubation, the increased β-catenin phosphorylation, along with its reduced protein levels (i.e., proteasomal degradation), was consistent with a temporary *Wnt* de-activation. In contrast, *Wnt* recruitment (i.e., increased β-catenin levels) occurred after 72 h of exposure, implying an anti-adipogenic action. Of note, beyond *Wnt*, 4-HNE treatment led to a time-sensitive and opposite activation of p38 (up-regulation persisting after 48 and 72 h), as well as the early recruitment (ie, after 24 h) of ERK1/2 and SAPK/JNKs, consistent with the antagonizing effects of these MAPKs’ signaling during various steps of lineage commitment and the differentiation of ASCs after exposure to oxidative stress [39,40]. In this regard, the current evidence suggests that, despite the controversial role of JNK in regulating stem cell lineage commitment and differentiation, ERK and p38 activity will be necessary to initiate the proliferative step (i.e., mitotic clonal expansion) of the precursor cells in the differentiation process [39,40]. However, thereafter, these signal transduction pathways need to be shut off, since their persistent activation hampers adipogenic differentiation via the inhibition of key metabolic adipocyte genes (i.e., C/EBP and PPAR-γ) [39,40]. The observation that 4-HNE simultaneously engages *Wnt* and MAPKs with opposite effects provides a novel and conceptually important breakthrough, allowing us to better understand the adipogenic program. We thus suggest that during the early steps of adipogenesis, a time frame in which *Wnt* is transiently de-activated (i.e., a pro-adipogenic signal), 4-HNE recruits both the mitogen (ERK1/2)- and the stress (JNK, p38)-sensitive kinases, providing a unique “point of convergence” between these signaling checkpoints in promoting ASC proliferation. However, the subsequent activation of Wnt, along with the persistent recruitment of MAPK signaling, may well explain the ability of 4-HNE to impair adipogenic commitment and differentiation after prolonged exposure (i.e., 48–72 h). On the other hand, since we did not check Wnt recruitment during the later stages of differentiation (i.e., at day 14), nor the expression of other “canonical” adipogenic markers (i.e., C/EBPα, PPAR-γ, adiponectin, etc.), we speculate that the ability of 4-HNE to hamper adipogenesis might likely be the result of a coordinated activation/inhibition of several (yet unexplored) signaling pathways within specific “time-windows”.

Our study has a few potential limitations and strengths that need to be briefly outlined. First, the relatively small sample size of the study population may have biased the results. However, the effect of a small sample size may have incurred a false negative (type II) error and not positive results. Nonetheless, the number of participants included in our work was adequately powered to detect the magnitude of difference in adipose 4HNE abundance observed between the groups, and it also closely matches that reported in other studies where the in-depth characterization of AT was performed [41,42,43]. Second, since the accumulation of HNE in some tissues increases with aging [44] and after hormonal changes (i.e., menopause) [45,46], one may argue that the older age and the menopausal status of the OBT2D patients might have also affected the results. On the other hand, in a multiple regression model (Table 2), the adipose cell size was the only variable that significantly predicted the adipose accumulation of 4-HNE, independent of age and sex (as confounding variables), suggesting that this molecule may well behave as a unique signature of “diabetic” fat, independent of ageing or menopause. Finally, the time-course monitoring of the key markers of adipogenic differentiation (i.e., C/EBPα, PPAR-γ, adiponectin, etc.) would have been required to further circumstantiate the ability of 4-HNE to hamper adipogenesis. These constraints have to be weighed against the strengths of our research, which reside in the use of primary cultures of human ASC lines, providing “physiological” levels of 4-HNE in a relevant concentration range. Moreover, a comprehensive and simultaneous evaluation of the key signaling checkpoints regulating ASCs’ commitment and differentiation provides important, breakthrough knowledge on human adipose biology.

In conclusion, the increased accumulation of 4-HNE in subcutaneous abdominal fat appears to characterize “diabetic” fat and detrimentally affects the adipogenic potential of adipose progenitor cells through the time-dependent activation of the *Wnt* and MAPK signaling cascades. Future studies on larger independent cohorts are warranted to better elucidate the mechanistic insights linking lipoxidation in adipose tissue to systemic insulin resistance and, possibly, offer novel therapeutic strategies for the treatment of adiposity-linked metabolic dysfunctions.

## 4. Materials and Methods

### 4.1. Subjects

We studied 14 volunteers recruited from those referred to the outpatient clinics of the Department of Endocrinology and Metabolism of Perugia Hospital. Four participants were obese and diagnosed with type 2 diabetes (OBT2D), while four lean and nondiabetic individuals represented the control group (CTRL).

The OBT2D patients were eligible for the study if they met the following criteria: (1) a BMI 30–39 kg/m^2^; (2) acceptable metabolic control presenting as HbA1c < 8.5%; (3) no significant micro- or macro-vascular complications; (4) a lack of concomitant endocrine diseases (i.e., thyroid dysfunctions, Addison’s disease, etc.); (5) weight stability during the 6 months preceding the study; (6) no ongoing treatment with insulin; and (7) no use of vitamin supplements. In addition, the CTRL subjects were enrolled if they displayed the following characteristics: (1) a healthy state, as determined by medical history, a physical examination, and screening laboratory evaluations; (2) a BMI 18–24 kg/m^2^; (3) normal physical activity and drinking habits; and (4) no current regular medication.

An informed written consent was obtained from all volunteers before their participation in the study, which has been approved by the Ethical Committee of Umbria Region (CEAS Protocol N. 5198/15/AV) and carried out according to Declaration of Helsinki principles.

#### 4.1.1. Body Composition and SCAAT Needle Biopsy

In the basal evaluation, a body composition analysis (dual-energy X-ray absorptiometry, DPX-IQ; Lunar Radiation, Madison, WI, USA) and a needle biopsy from the sc abdominal AT (SCAAT) were performed [47]. The AT specimens (1–2 g) were immediately processed for cell isolation and fat cell size measurement [48] and partly homogenized for the detection of protein–HNE adducts (see below).

#### 4.1.2. ASC Isolation, Cell Viability, and Adipogenic Differentiation

ASC isolation was performed as previously reported [17,49]. Since lipid peroxidation products are more “toxic” when cells are exponentially grown [50], the experiments for assessing cell viability were carried out on resting, “synchronized” ASCs. In detail, before treatment, cell cultures were starved through 48 h of incubation in a washout DMEM-F12 medium containing 0.5% FBS and antibiotics. The cells were left “untreated” (control) or incubated with 4-HNE (JAICA, Japan Institute for the Control of Aging, Nikken SEIL Co., Ltd., Fukuroi, Shizuoka, Japan) at concentrations between 0.05 and 50 μM for 24, 48, and 72 h, respectively. For the cell treatment, 4-HNE was dissolved in ethanol, and the final concentration of ethanol in the culture medium did not exceed 0.1% (*v*/*v*). Equivalent quantities of ethanol were also added to the control cells.

Cell viability was examined using a (3-(4,5-dimethylthiazolyl-2)-2,5-diphenyltetrazolium bromide) (MTT) reduction test (Sigma-Aldrich, Saint Louis, MO, USA), which evaluates mitochondrial activity, cellular growth, and survival [51]. Absorbance at a wavelength of 570 nm was measured with a microplate reader (DTX 880 Multimode Detector Beckman Coulter, Bergen County Technical Schools Stem Cell Lab, New York, NY, USA).

For differentiation into mature adipocytes, the ASCs were treated with a plating medium (without serum) enriched with DMI (100 nM dexamethasone, 540 μM isobutylmethylxanthine (IBMX); 500 nM human insulin), 200 pM triiodothyronine, 10 μM pioglitazone, and antibiotics. IBMX was removed after 2 days, and the differentiation medium was changed every 2 days for 2–4 weeks. To evaluate the effects on ASC differentiation, 4-HNE was added at the final concentration of 1 μM to the differentiation mixture and to the cell culture media during differentiation, as indicated. The differentiation of ASCs into lipid-filled cells with an adipocyte morphology was monitored morphologically. Undifferentiated cells were considered differentiated when their cytoplasm was completely filled with multiple fat droplets visible as doubly refractile inclusions via low-power phase-contrast microscopy. The lipid nature of these inclusions was further confirmed through Oil Red O (ORO) staining. The level of ORO uptake was determined after careful isopropanol extraction by measuring the optical density (OD) of the extracted dye at 510 nm [17].

#### 4.1.3. Western Blots

Whole-cell protein and whole-AT lysates were prepared, and 20 µg of protein was fractionated using 10–12% SDS-PAGE and transferred to a nitrocellulose membrane (Millipore), as previously described [17]. The protein–HNE analysis was carried out using an anti-HNE-adduct antibody (clone HNEJ-2; JAICA) which shows a high affinity for the HNE–histidine adduct and almost negligible reactivity with proteins treated with other aldehydes (i.e., 2-nonenal, 2-hexenal) [52]. For the other immunoblots, the following antibodies were used: β-catenin (BD Biosciences, San Jose, CA, USA); extracellular-signal-regulated kinases (ERK1/2) (#4695); phosphorylated ERK1/2 (Thr202/Tyr204) (#4377), stress-activated protein kinases/Jun amino-terminal kinases (SAPK/JNK) (#9258); and phosphorylated SAPK/JNK (Thr183/Tyr185) (#4671), GAPDH (#5174), α-tubulin (#2144), phospho-β-catenin (Ser33/37/Thr41) (#9561), phospho-p38 (#4511), anti-rabbit IgG HRP-linked antibody (#7074), and anti-mouse IgG HRP-linked antibody (Cell Signaling Technology, Beverly, MA, USA). The immunoreactive signals were detected using an enhanced chemiluminescence system (Pierce ECL Plus, Thermo Scientifics, Rockford, IL, USA).

For all the Western blot analyses, each sample of lysate obtained from either the whole AT or ASCs of the donors was analyzed at least three times for each experiment.

#### 4.1.4. Intracellular ROS Production and Cell Viability

Intracellular ROS production was evaluated using the oxidation-sensitive fluorescent probe 2′,7′-dichlorofluorescein diacetate (H_2_DCF-DA) (Sigma Aldrich, Saint Louis, MO, USA) [53]. The data were expressed as folds of optical density (OD) adjusted for the viable cells. Cell viability was examined using a 3-(4,5-dimethylthiazolyl-2)-2,5-diphenyltetrazolium bromide (MTT) reduction test [51] (Sigma-Aldrich, Saint Loius, MO, USA). This assay is based on the ability of live cells to convert tetrazolium salt into purple formazan. Absorbance at a wavelength of 540 nm was measured with a microplate reader (DTX 880 Multimode Detector Beckman Coulter).

#### 4.1.5. Statistics

The statistical comparisons of the groups were carried out through Student’s *t*-test and the Mann–Whitney U-test for parametric and non-parametric (Kolmogorov–Smirnov test) variables, respectively. Univariate correlations were assessed using Spearman’s rho (rs) and partial correlation, while multivariate relationships were evaluated using multiple regression models. Variations in cell viability over time were analyzed using three-way repeated-measure ANOVA, with Geisser–Greenhouse adjustments for non-sphericity and Bonferroni’s post hoc test where appropriate. For the calculation of IC50, the survival data were evaluated through variable slope curve fitting with GraphPad Prism (GraphPad software, CA). The data are shown as the mean ±SEM or median (25th; 75th percentile). *p* Values < 0.05 were regarded as statistically significant. The statistical analysis was carried out using the NCSS software (Kaysville, UT, USA).

## Figures and Tables

**Figure 1 ijms-24-16645-f001:**
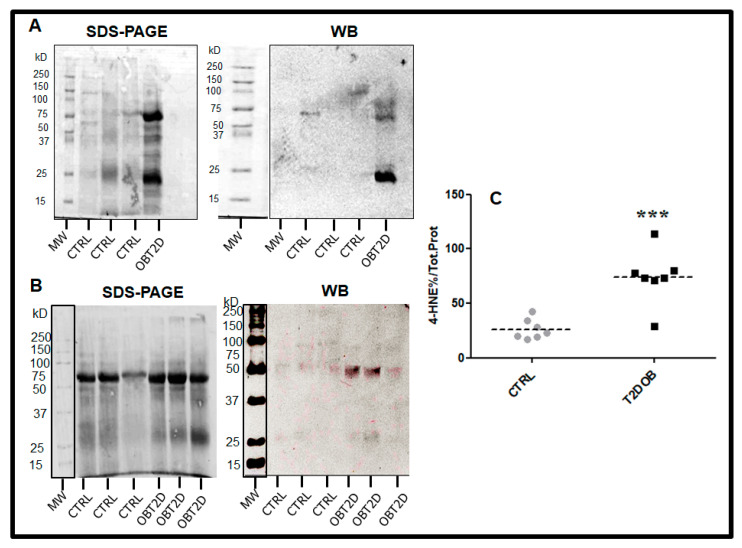
Panel (**A**,**B**): Representative blots of 4-HNE–protein adducts in adipose tissue of six control subjects (CTRL) and four obese patients with type 2 diabetes (OBT2D). MW: molecular weight; kD: kilodalton. SDS-PAGE: Sodium dodecyl-sulfate polyacrylamide gel electrophoresis; WB: Western Blot. Panel (**C**): Dot plot graph showing the relative optical density of 4-HNE–protein adducts expressed as percentage of total protein in AT of CTRL subjects (grey circles) and OBT2D patients (dark squares). Dotted lines represent the mean values in each group (*** *p* < 0.001 vs. CTRL).

**Figure 2 ijms-24-16645-f002:**
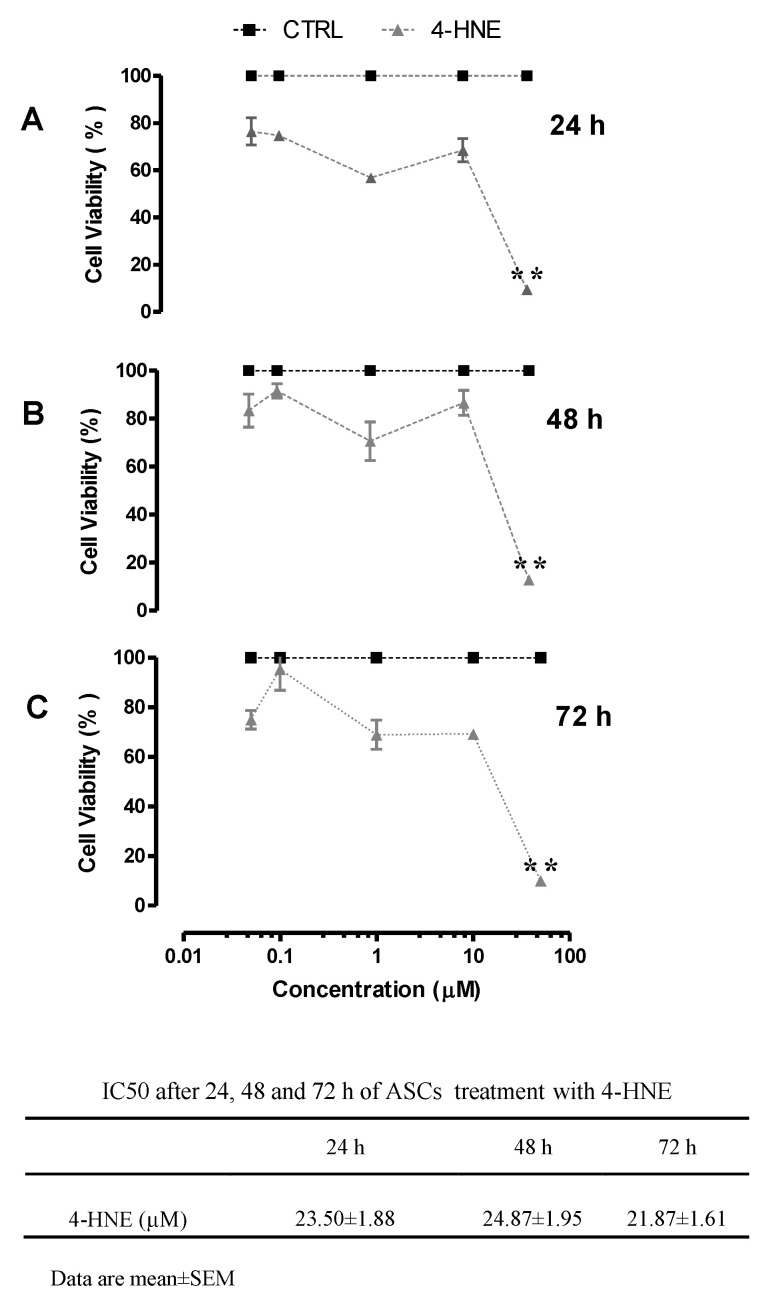
Cell viability of human ASCs isolated from CTRL subjects (n = 4) after 24 h (panel (**A**)), 48 h (panel (**B**)), and 72 h (panel (**C**)) of treatment with increasing concentrations (0.01, 0.1, 1, 10, and 50 µM) of 4-HNE, as determined using the MTT reduction test. OD: optical density. Data are expressed as the percentage of viable cells relative to the control (CTRL), which was considered as 100%, and represent the mean ± SEM of at least three different experiments (sample assayed three times for each experiment). ** *p* < 0.01 vs. control (untreated cells) based on 3-way repeated-measure ANOVA.

**Figure 3 ijms-24-16645-f003:**
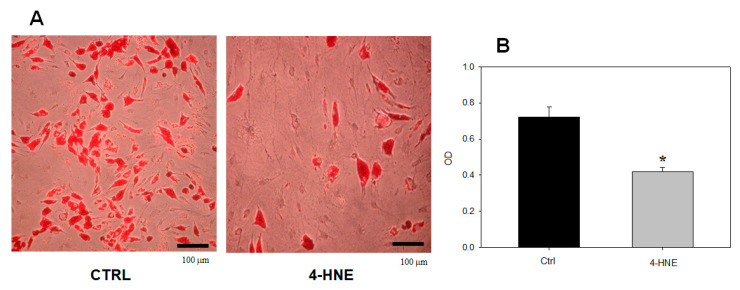
Oil Red O staining (panel (**A**)) and optical density (OD) (panel (**B**)) of human ASCs isolated from control subjects (n = 4) and treated with 1 µM of 4-HNE after the induction of differentiation. Magnification of representative microscopic images was 10X. * *p* < 0.01 vs. control (CTRL; untreated cells).

**Figure 4 ijms-24-16645-f004:**
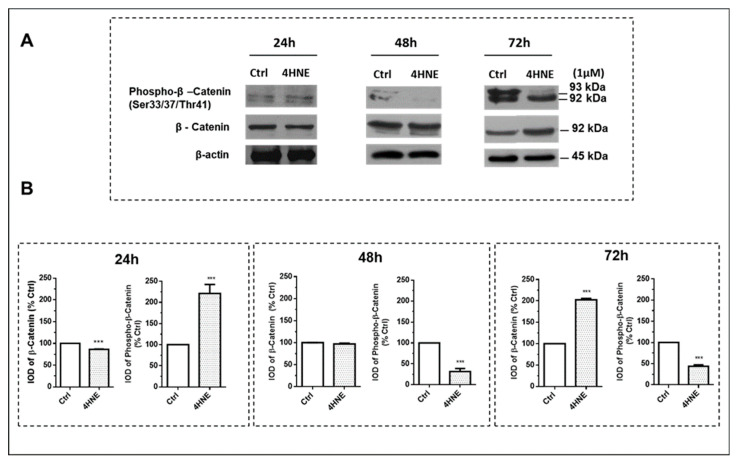
Recruitment of WNT signaling in human ASCs isolated from CTRL subjects (n = 4) treated with 4-HNE. The effect on cellular β-catenin and its phosphorylation (phospho-β-catenin) was investigated after challenging the cells with 4-HNE (1 µM) and immunoblots obtained after 24, 48, and 72 h of incubation (panel (**A**)). Data are expressed as the protein percentage in the cell treatment relative to control tests (CTRL), which were considered as 100%, and represent mean ± SEM of at least three different experiments (sample assayed three times for each experiment) (panel (**B**)). β-Actin was used as the loading control. IOD: integrated optical density; *** *p* < 0.05 vs. CTRL.

**Figure 5 ijms-24-16645-f005:**
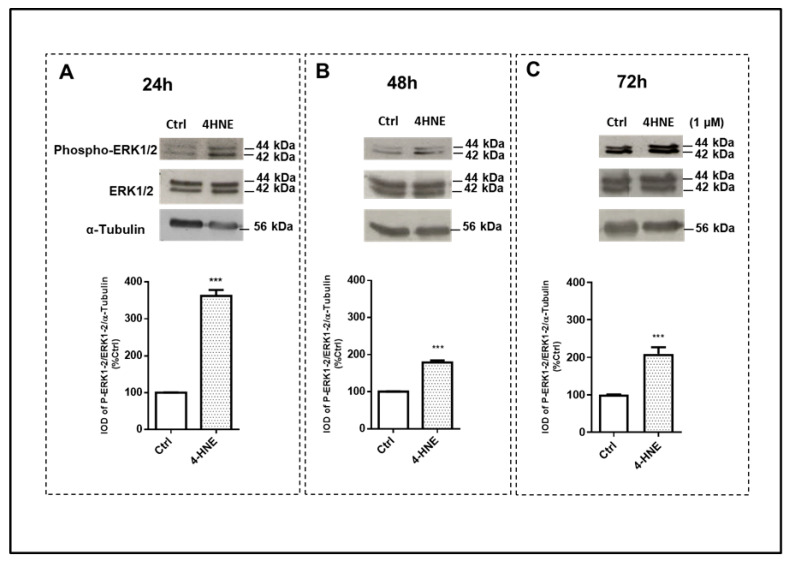
Immunoblot analysis of extracellular-signal-regulated kinases (ERK 1/2) and its phosphorylation (phospho-ERK1/2) after 24 h (panel (**A**)), 48 h (panel (**B**)), and 72 h (panel (**C**)) of treatment with 4-HNE (1 μM) in human ASCs isolated from CTRL subjects (n = 4). Data were calculated based on the ratio of the integrated optical density (IOD) of the phosphorylated protein to that of total protein and that of the housekeeping protein (α-Tubulin). Results are expressed as the percentage of the cell treatments relative to the control tests (CTRL) that were considered as 100% and represent the mean ± SEM of at least three different experiments (sample assayed three times for each experiment). α-Tubulin was used as the loading control. *** *p* < 0.001 vs. CTRL.

**Figure 6 ijms-24-16645-f006:**
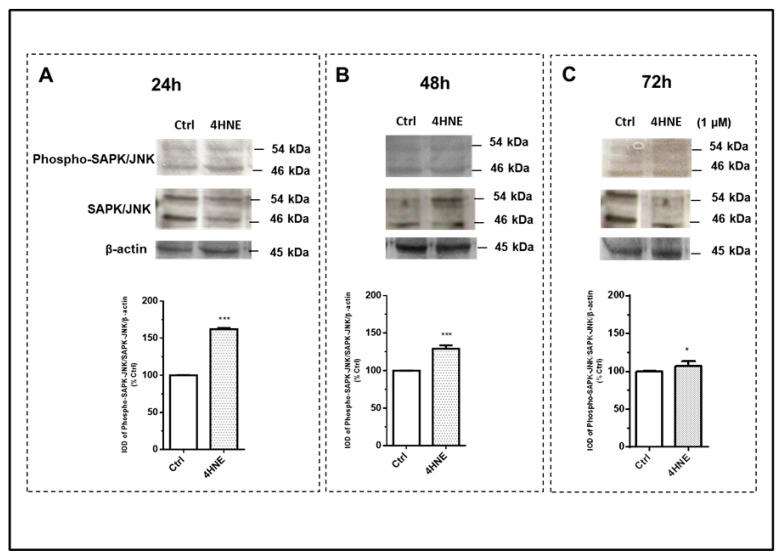
Immunoblot analysis of stress-activated protein kinases/c-Jun N-terminal kinases (SAPK/JNK) and their phosphorylation (phospho-SAPK/JNK) after 24 h (panel (**A**)), 48 h (panel (**B**)), and 72 h (panel (**C**)) of treatment with 4-HNE (1 μM) in human ASCs isolated from CTRL subjects (n = 4). Data were calculated based on the ratio of the integrated optical density (IOD) of the phosphorylated protein to that of total protein and that of the housekeeping protein (β-actin). Results are expressed as the percentage of the cell treatments relative to the control tests (CTRL), which were considered as 100%, and represent mean ± SEM of at least three different experiments (sample assayed three times for each experiment). *** *p* < 0.001 vs. CTRL. * *p* < 0.05 vs. CTRL.

**Figure 7 ijms-24-16645-f007:**
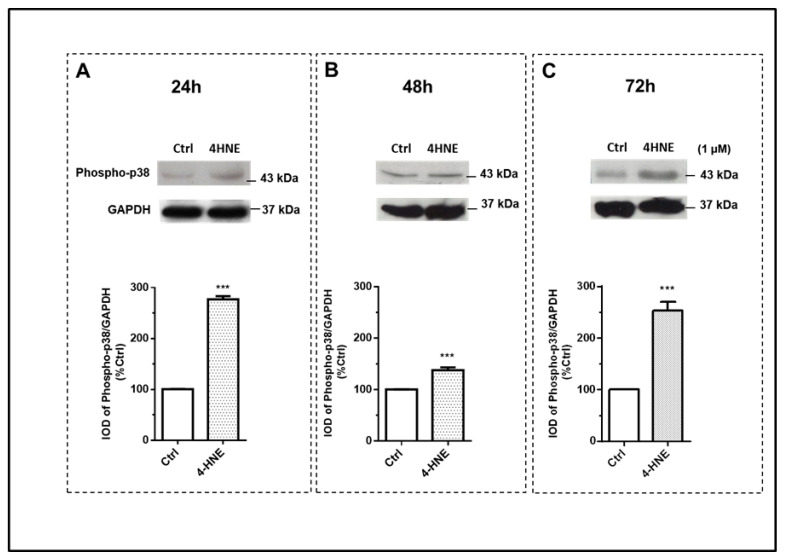
Immunoblot analysis of p38 MAPK and its phosphorylation (phospho-p38) after 24 h (panel (**A**)), 48 h (panel (**B**)), and 72 h (panel (**C**)) of treatment with 4-HNE (1 μM) in human ASCs isolated from CTRL subjects (n= 4). Data were calculated based on the ratio of the integrated optical density (IOD) of the phosphorylated protein to that of the housekeeping protein (Glyceraldehyde 3-phosphate dehydrogenase, GAPDH). Results are expressed as the percentage of the cell treatments relative to the control tests (Ctrl), which were considered as 100%, and represent the mean ± SEM of at least three different experiments (sample assayed three times for each experiment). *** *p* < 0.05 vs. CTRL.

**Table 1 ijms-24-16645-t001:** Main characteristics of the study subjects.

	CTRL	OBT2D	*p* Value
n	7	7	
Sex (M/F)	4/3	5/2	
Age (yr)	39 ± 3.5	60 ± 3.8	<0.01
BMI (kg/m^2^)	23.2 ± 0.9	35.6 ± 1.9	<0.0001
Waist circumference (cm)	79.1 ± 3.14	109.1 ± 3.8	<0.001
WHR	0.86 ± 0.04	1.2 ± 0.09	<0.001
Total FM (kg)	14.1 ± 2.6	45.3 ± 3.1	<0.001
Trunk FM (kg)	8.4 ± 0.7	22.5 ± 1.1	<0.001
FFM (kg)	58.2 ± 2.1	65.9 ± 2.3	0.01
HbA1c (%)	4.6 ± 0.4	7.7 ± 0.7	<0.001
Adipocyte cell diameter (μm)	70.5 ± 3.2	105 ± 1	<0.001

Data are mean ± SEM. FM, fat mass; FFM, fat-free mass. WHR: waist-to-hip ratio.

**Table 2 ijms-24-16645-t002:** Multiple regression analysis for the prediction of adipose 4-HNE abundance in the whole study cohort.

Independent Variables	β	*p* Value	Adjusted R^2^
Adipocyte cell diameter	0.807	<0.001 <0.001	0.622
Sex	−0.109	0.889	
Age	−0.045	0.858	

## Data Availability

The data are available on request from the authors.

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
