# Peer review of "Accumulation of 4-Hydroxynonenal Characterizes Diabetic Fat and Modulates Adipogenic Differentiation of Adipose Precursor Cells"

_ijms, 2023, doi:10.3390/ijms242316645_

Round 1

Reviewer 1 Report (New Reviewer)

Comments and Suggestions for Authors

In this article, the authors aim to demonstrate how the accumulation of 4-hydroxynonenal (4-HNE) is present in the fatty cells of obese individuals affected by type 2 diabetes and how this compound, produced by the oxidation of lipid membranes, is involved in the mechanisms of adipogenic differentiation.

The aim of the study is interesting however, the authors need to improve the quality and completeness of the data collected, starting with the numerosity of the starting sample.
A cohort of only 4 controls and 4 patients is too small to give the results clinical relevance. In addition, the age difference between the groups is too large to allow effective comparison.
Last, the authors have not effectively demonstrated how 4-HNE is co-involved in adipogenesis. To this end, it is necessary to perform real-time PCRs by monitoring the variation in expression of the main markers involved in this process (i.e., early adipocytes: ADD1, C/EBPβ, KLF5, PPARγ, LPL, leptin and adiponectin; mature adipocytes: PPARγ, C/EBPα, adiponectin, adipsin, adipocyte protein 2, and P2Y11).

Others:
1. Figure 3 is not mentioned in section 2.1.1. Check for correct insertion and numbering of figures.
2. Evaluate the protein expression of ERK1/2, c-Jun, and p38 by calculating the ratio of the phosphorylated fraction to the total fraction to verify the actual protein activation.

Author Response

Reviewer 2 Report (New Reviewer)

Comments and Suggestions for Authors

In this paper the authors attempted to test their hypothesis that 4-HNE might act as a molecular instigator of blunted ASCs differentiation.  Eight subjects were included in the study that has two groups: OBT2D and CTRL subjects. The Protein-HNE adducts in subcutaneous abdominal AT (SCAAT) biopsy assessed by western blot. The effects of 4-HNE were then studied on primary  cultures of ASCs focusing on cell viability, adipogenic differentiation, “canonical” Wnt and MAPKs 23 signaling pathways. There are some concerns with this paper that need to be addressed.

Major concerns:

1)      The ages of the two groups of subjects are too different: the OBT2D group (58+/- 5.1 years old) and CTRL group (32+/-3.8 years old). The difference is too big for these two groups to be compared to each other.  The CTRL group can be considered young adult and all individuals in the OBT2D group should have past menopause age for women and could be considered old for man.

There are too many contributing factors, such as hormonal changes and aging processes, for differences seen in these two populations. Oxidative stress occurs naturally and plays an important role in the aging process. The oxidative stress markers have been reported to increase in older adults. Menopause is associated with an increase in oxidative stress (Menopause 201219, 361–367; Gynecol. Endocrinol. 201632, 982–985.).

For the authors to make the comparison of the OBT2D and CTRL groups, subjects with similar age need to be included in the study.

2)      The n number is on the smaller side for each group. Age matched groups need to be included with the study with minimum of n=5 to each group.

3)      For western blot analysis, an internal protein (housekeeping) control should be included to account for the variations seen with loading and western blot procedure. Please show more samples for the OBT2D group as well, n=3-4 for both groups.

Specific concerns:

1)      Line 64: “appear very sensitive and vulnerable.”

Should that be:  appear to be very sensitive and vulnerable…?

2)      Line 78: “to test the hypothesis whether…”

Should that be to test the hypothesis that…?

3)      For figure 1, please show more samples for the OBT2D group as well. The same as CTRL group, please show n=3-4 for both groups.

4)      For the cell analysis (figures 4-7), not sure why the same sample is run three times. Does it mean that triplicate was performed for each treatment and the experiment was performed three times.

Comments on the Quality of English Language

The paper would benefit from some professional language editing so that the readers can understand what the authors is trying to convey. 

Author Response

Reviewer 3 Report (New Reviewer)

Comments and Suggestions for Authors

This is an extremely valuable study that was conducted using human samples. But unfortunately, this manuscript cannot be accepted because there is a major problem with the experimental design. 

There are major problems with the research plan.

1. The difference in age between the control group and the diabetic group was significant.

2. There were two men and two women in both the control group and the diabetic group. There are large gender differences in glucose tolerance and fat metabolism in premenopausal women. Therefore, the influence of estrogen cannot be ignored in the analysis of the two women in the control group. 

3. If the authors intended to analyze four people in the control group and four people with diabetes, they should match for gender, such as including only men.

4. The ages of the control group and the diabetic group should be as similar as possible. The age difference between the control group and the diabetic group in the results of this study was too large to exclude that the difference was caused by the effects of aging.

I think Figure 2 on line 103 is a typo, and Figure 3 is correct.

Author Response

Plase, see the attached file.

Round 2

Reviewer 1 Report (New Reviewer)

Comments and Suggestions for Authors

Although the revised paper has numerous improvements, such as expanding the case study, it needs further experimentation:

1. The authors have justified both in the cover letters and in the discussion the choice to evaluate exclusively by WB the canonical Wnt and the MAPKs pathways but the integration of the data with time-course monitoring of key markers of adipogenic differentiation is a strong confirmation of their hypothesis. I, therefore, reiterate what was said above.

2. Figure 3 is acquired in different ways since the difference in background between the two is evident. It is required to show that both images were acquired under the same conditions.

3. Although the caption of Figure 7 states "Data were calculated by the ratio of the integrated optical density (IOD) of the phosphorylated protein to that of total protein and that of the housekeeping protein (Glyceraldehyde 3-phosphate dehydrogenase; GAPDH);" the figure was not correctly updated.

Author Response

Please, for reply, find the file herein attached.

Reviewer 2 Report (New Reviewer)

Comments and Suggestions for Authors

The revision has significant improvement. However, there are still many very critical details missing from the the description of the experiments and the presentation of the results. 

1) It is still not clear what the "n" is for the experiment that is presented in Figures 2, and 4-7? Are all the ASCs cells isolated from one donor?  Please specify the number of donors from which the ASCs isolation has been performed? 

2) For figure 3, the background for the CTRL and 4-HNE treated cells are way different. Please make sure these two pictures are taken under the same condition. For the B panel of figure 3 , when performing the isopropanal extraction, how many samples were analyzed and how many cells were in each sample? As it was shown in figure 2, the ASCs viability decreased with  4-HNE treatment. Therefore, the decrease in OD510 measured could represent the less adipocyte cells rather than less lipid accumulation in the cells. It would be better to have the OD510 reading normalize with cell number. 

3) Differentiated adipocytes should have lipid droplets that are visible under microscope, however, the picture presented in figure 3 does not seem to have visible lipid droplets. Why is that? Could you please provide a picture with higher magnification? or at least have a scale bar for the picture presented in figure 3. 

Comments on the Quality of English Language

It is still not clear what the "n" is for the experiment that is presented in Figures 2, and 4-7?  The description by the authors are confusing.

See comments above. 

Author Response

Please,  find herein with the file attached with for replies.

Reviewer 3 Report (New Reviewer)

Comments and Suggestions for Authors

The authors sincerely respond to reviewers' questions. Although it is not completely sufficient, I will accept it this time.

Author Response

We are grateful to Reviewer 3 for his/her suggestions and criticisms which have improved the quality of our work.

Round 3

Reviewer 1 Report (New Reviewer)

Comments and Suggestions for Authors

The authors corrected all previous critical issues.

Author Response

We are grateful to the Reviewer 1 for his/her valuable comments and suggestion to improve the quality of our work.

Best regards.

Giuseppe Murdolo

encl. 

Reviewer 2 Report (New Reviewer)

Comments and Suggestions for Authors

Please add a scale bar to each of the picture presented in the paper. 

Author Response

We are grateful to the Reviewer 2 for his/her criticism. Please, find enclosed the answer to the rebuttal.

Best regards.

Giuseppe Murdolo

encl.

This manuscript is a resubmission of an earlier submission. The following is a list of the peer review reports and author responses from that submission.

Round 1

Reviewer 1 Report

Comments and Suggestions for Authors

4-hydroxynonenal (4-HNE) is an oxidation product of w-6 fatty acids. Levels of this lipid molecule in skeletal muscle and adipose tissue are correlated with obesity and insulin resistance. The authors studied the levels of 4-HNE in the subcutaneous adipose tissue of obese individuals with type 2 diabetes (OBT2D) and compared the results to control individuals. They further studied the effect of 4-HNE on adipocyte stem cell viability, differentiation, and signaling.

The data in the current manuscript is not convincing regarding the levels of 4-HNE or its effect on adipocyte stem cell biology for the reasons given below:

How was the purity of the ASCs ascertained?

Line 87 – please include data showing the larger adipose cell size in OBT2D individuals compared with CTRL individuals.

Line 92 – include data showing the association of levels of 4-HNE with the cell size.

Figure 1 – show data for each sample (individual). There are only 4 OBT2D and 4 CTRL samples. The same should be done for all the other data. How is the data shown in the graph calculated?

Line 104 – include data showing effect of 4HNE on cell viability. How is viability affected by treatment for 24, 48, and 72 h?

Line 123 – the authors state that after 48 h…catenin phosphorylation increased. However, the data shows a decrease. There is almost no P-catenin band.

Line 126 – please do not state results without showing the data.

Figure 2 – how many times was this experiment repeated? And how was it repeated? Are the graphs showing means ± SEM? How was statistical significance calculated?

Figures 3 to 6 – the authors state that the data are means ± SEM of at least three different experiments. What does that mean? Was the sample collected at least three times from each participant? Were the ASCs collected at least three times from each participant? Were the ASCs set up at least three times for each experiment? Or was each sample assayed at least three times?

Figure 5 – the blots for the P-JNK are not convincing and should be redone.

Discussion, Line 81 – the sentence is ambiguous. Given that the role of ROS in this study is unclear, the authors should clearly state that only high non-physiological concentrations of (50 uM) of 4-HNE induced an increase in ROS production.

Discussion, line 108 – the authors are concluding that their data showing reduced levels of catenin in 4HNE treated cells is consistent with Wnt deactivation. This is inaccurate and not supported by their data. Furthermore, their data is the opposite to the data from many other labs show that inactivation of Wnt is a prerequisite to differentiation of preadipocytes to adipocytes. It is also the opposite of their hypothesis.

Discussion, line 117 -119 – this conclusion is not supported by their data. See comment above.

Reviewer 2 Report

Comments and Suggestions for Authors

The author investigated accumulation of 4-hydroxynonenal characterizes the diabetic fat and modulates adipogenic differentiation of adipose precursor cells. The topic is interesting. However, the experimental design of this study has major defects, and I have listed my comments below.

1.       The biggest drawback of this study is that the sample size of the population is too small. Due to the huge individual differences of the population, the research results of n=4 are not convincing enough. The credibility of the experiment is poor. If the author cannot expand the number of samples, I cannot recommend the publication of this manuscript.

2.       Line 76-81. The space between lines seems to be inconsistent with the previous paragraph, please modify it.

3.       Line 93. It should be R2=0.58.

4.       Figure 3A. Theoretically, there should be three bands of three phosphorylation sites, but it seems that the author provided it is not so.

5.       Figure 3A, 4A, 5A. Please provide the molecular weight of different bands (kDa).

6.       Figure 3B, 4B, 5B. Why the CTRL group does not have error bar.

7.       Figure S1. Please provide the magnification or scale of the picture.

8.       Figure S2. Is each sample done only once? Is there any repetition?

9.       ijms-2564592-original-images. The strips here are still spliced, not the original strips.

Comments on the Quality of English Language

Minor editing of English language required.

Round 2

Reviewer 1 Report

Comments and Suggestions for Authors

The authors have addressed some of this reviewer’s comments but not all. Please address the following comments, which were also in my previous review:

They have assayed only 4 OBT2D and 4 CTRL samples and should show the data for reach sample. These experiments are already done. Each sample was assayed three times, which is not an optimal way to check for reproducibility. Showing data for each sample will make the results stronger.  

The authors should submit the original blots; not cropped or modified in any way and should include the size marker. They have submitted a file titled “Original images”. However, this file includes the same figures that are in the manuscript.

In the Abstract, Results section, the authors state that, “4-HNE hampered adipogenic ASCs differentiation by a timely-regulated activation of Wnt/β-catenin, p38MAPK, ERK1/2- and JNK-mediated pathways”. In Supplemental Fig. 1, they show that 4-HNE treated cells at day 14 have lower levels of Oil Red O staining than the control cells. They show increased activation of catenin after 24 h of incubation with 4-HNE but not after 48 and 72 h. Can results obtained at D1 be extrapolated to results at D14? It seems like a stretch to this reviewer.

Why is the data showing the effect of 4-HNE in the Supplemental? Is Oil Red O staining sufficient to conclude decreases differentiation of ASCs?

Reviewer 2 Report

Comments and Suggestions for Authors

The authors have done their best to answer my questions and improve the quality of the manuscript.

Author Response

Manuscript # IJMS-2564592 (Round 2)

"Accumulation of 4-hydroxynonenal characterizes the diabetic fat and modulates adipogenic differentiation of adipose pre-cursor cells"

 Reviewer 2

Reviewer #2: We sincerely express our gratitude to he reviewer #2 for the effort and the consideration for our work.

Round 3

Reviewer 1 Report

Comments and Suggestions for Authors

The authors have failed to address the following comments, which were in both first and second reviews.

They have assayed only 4 OBT2D and 4 CTRL samples and should show the data for reach sample. This does not require new experiments. Each sample was assayed three times, which is not an optimal way to check for reproducibility. Showing data for each sample will make the results stronger.  

The authors should submit the original blots; not cropped or modified in any way and should include the size marker. They have submitted a file titled “Original images”. However, this file includes the same figures that are in the manuscript.
